# Improving Antimicrobial Stewardship in Acute Sore Throat: Comparison of FeverPAIN and McIsaac Scores with Molecular Point of Care Testing Using Abbott ID NOW

**DOI:** 10.3390/diagnostics14232680

**Published:** 2024-11-27

**Authors:** Rob Daniels, Tarek El Omda, Kinan Mokbel

**Affiliations:** 1Faculty of Health Care Professions, St Luke’s Campus, University of Exeter, Exeter EX1 2LU, UK; k.a.mokbel@exeter.ac.uk; 2TASC Primary Care Network, Townsend House Medical Centre, Seaton EX12 2RY, UK; tarek.elomda@nhs.net

**Keywords:** point of care testing, group A streptococcal (GAS) infections, quality improvement, antimicrobial stewardship

## Abstract

**Background:** The diagnosis of streptococcal throat infection is an area where current practice results in significant over-diagnosis of bacterial infection, with the resulting implications for antimicrobial use and resistance. The use of molecular point of care testing (POCT) has previously been shown to alter antibiotic prescribing decisions when compared to Centor scoring. This paper explores the impact of the addition of POCT to clinical assessment using the McIsaac and FeverPAIN scoring systems. **Methods:** Retrospective analysis of the clinical records of 144 patients who had previously received point of care testing as part of a trial of POCT as an adjunct to usual care was undertaken. The McIsaac and FeverPAIN scores were calculated and then compared with the POCT result originally obtained. The records were also reviewed to identify further consultations within 10 days of being tested. **Results:** Sufficient clinical data were available to calculate the FeverPAIN and McIsaac scores in 81% of patients and outcome data were available for 96.5% of patients. For patients with a FeverPAIN score of 4 or 5, 50–67% of patients had a positive POCT result, while for patients with a McIsaac score of 4 or 5, 50–68% had a positive POCT result. Moreover, 16.7% of patients who tested negative, and 16.3% of patients who tested positive, had a further consultation within 10 days of being assessed. **Conclusions:** Although relatively few patients in this study had maximum scores on the FeverPAIN and McIsaac scoring, the addition of POCT was shown to alter antibiotic prescribing decisions in a significant number of patients, supporting the use of Abbott ID NOW point of care testing to reduce antibiotic prescribing. Larger studies are required to confirm these results and explore the health economic aspects and potential impacts on health inequalities.

## 1. Introduction

Infections with streptococci continue to cause significant morbidity and mortality in developed and developing nations. The most common presentation of GAS is acute sore throat, which remains one of the most frequent indications for antibiotic prescribing, with an estimated five prescriptions per 100 patients per year in Europe [1]. Epidemiological studies suggest that only 15–30% of sore throats are caused by bacterial pathogens [2], but antibiotics are prescribed in 60–70% of cases [3]. NHS England prescribing data [4] identified 2.6 million prescriptions for phenoxymethylpenicillin between April 23 and March 24, the majority of which were for sore throat. There are many reasons for this, with anxiety around rare but serious complications of streptococcal infection and perceived patient expectations being prominent, together with clinical uncertainty in terms of diagnosis. The management of acute sore throat also presents logistical challenges to supply chains, as seen in the winter 2021/2022, when a significant increase in cases of streptococcal infections in the community [5] caused unprecedented demand for primary care services in the United Kingdom, causing shortages of antibiotic supply [6] and capacity issues in primary care.

Considerable clinical overlap exists in the symptoms and signs of streptococcal and viral throat infections, a situation further complicated by an asymptomatic carriage rate of 10.5% in young children [7]. Conventional microbiological microscopy, culture and antibiotic susceptibility testing can take up to 72 h, with earlier rapid antigen point of care tests having shown limited use due to inadequate sensitivity, around 70% for the test used in a previous study of the addition of rapid tests to clinical prediction rules [8]. The difficulty in accurately diagnosing or excluding streptococcal throat infections at the first point of contact has prompted the development of a range of clinical prediction rules, with the most commonly used being FeverPAIN [8] and Centor/McIsaac [9]. These combine symptoms and signs to stratify patients into groups based on the likelihood of identifying streptococci on culture. The current NICE guidance [10] in use in the UK, and guidance elsewhere such as the DEGAM guidance used in Germany [11], recommend the use of these tools in the assessment of sore throat, with prescribing decisions based on the result. The main difference between the current NICE guidance and the DEGAM guidance is that in the DEGAM guidance, rapid tests are suggested as an option for 3–15 year olds. The NICE guidance is summarised below in Table 1 and the DEGAM guidance in Table 2.

Although the use of clinical prediction rules is more accurate than clinical assessment alone [12], a FeverPAIN score ≥ 4 is still only associated with a probability of identifying streptococcal rather than viral infection in 62–65% of cases [13], so if antibiotics are used, this will still result in over-treatment in around 35% of patients. The TOAST study [14], a prospective study comparing clinical prediction rule (CPR) results with microbiological testing, confirmed the performance of this diagnostic and management strategy in clinical practice. Although the NICE guidance suggests consideration of antibiotics rather than automatic prescribing, it is the authors’ experience that in the majority of patients with the highest scores, patients will opt for a prescribing strategy in shared decision-making scenarios.

The advent of affordable, rapid, bedside testing for streptococcal infections using nucleic acid technology with high sensitivity and specificity has the potential to make a dramatic difference to antibiotic prescribing decisions whilst also reassuring patients and clinicians. The Abbott ID NOW system has a reported sensitivity of 98.5% and specificity of 93.4% [15], with results available in 3–6 min. A prospective quality improvement project looking at the impact of the addition of point of care testing (POCT) to clinical assessment with Centor scoring was previously carried out in primary care [16], which identified the potential to reduce antibiotic prescribing significantly through this strategy. Centor scores have received some criticism for their accuracy, so to further explore the potential of this approach, we compared the performance of McIsaac and FeverPAIN with the results obtained through nucleic acid testing with the Abbott ID NOW platform.

## 2. Method

This paper describes the secondary analysis of data collected during a quality improvement project to assess the impact of the addition of point of care testing to usual clinical care (using the NICE guidance [10]) in the assessment and management of acute sore throat. This original project was described by Daniels et al. [16]. In the initial project, patients presenting to GP practices or a community pharmacy between 1st November 2023 and 29th February 2024 complaining of acute sore throat were assessed according to the NICE guidance using Centor scoring. Patients who scored 2 or above were offered molecular point of care testing to guide antibiotic prescribing decisions. Patients who tested negative on the POCT were given symptomatic and safety-netting advice, while those who tested positive were treated with antibiotics according to local guidelines.

The process of care is described below in Figure 1.

The initial results [16] suggested that the addition of POCT to the use of the Centor clinical prediction rule would significantly reduce antibiotic prescribing. To determine if this applied to the FeverPAIN and McIsaac clinical prediction rules, a retrospective analysis of clinical records was undertaken by an independent clinician who was not involved in the previous project. Data were recorded on an Excel [17] spreadsheet and analysed in Stata [18].

To identify the clinical outcomes following POCT assessment, primary care records were analysed. Patients presenting to urgent or primary care in the United Kingdom can access a range of services, including emergency departments at local hospitals, minor injury units, out of hours primary care services and community pharmacies. Treatment summaries for each consultation are sent to the primary care provider, making the primary care electronic health record the most accurate database for further care episodes. Patient records were identified by a computerised electronic health record search. These patient records were reviewed, and where possible, the FeverPAIN and McIsaac scores calculated, depending on the clinical information contained in the records. These results were then compared with the result obtained through POCT and comparative data produced. In addition, medical records were reviewed to look for re-presentations in all the patients assessed during this project, where records were available. Unfortunately, it was not possible to accurately identify patients with acute sore throat who were not offered POCT due to inconsistent coding. Patients who presented to the community pharmacy were excluded from the retrospective FeverPAIN and McIsaac calculation due to the lack of sufficient clinical data in the pharmacy electronic health records, but their GP record was analysed for re-presentations to primary, urgent and secondary care providers.

## 3. Results

(i)Comparison of clinical prediction rules and POC test result

A total of 144 patients were assessed in the original project.

Moreover, 21 patients were assessed in a community pharmacy in the original project and their clinical records were unavailable for secondary analysis.

In 121 (84%) patients’ records, it was possible to calculate the FeverPAIN score, and in 123 (85%) patients’ records, the McIsaac scores were calculated.

The results for each CPR are shown below in Table 3 and Table 4.

(ii)Repeat consultation rates for patients receiving POCT

A total of 139 patient records were reviewed to identify return consultations within 10 days (five patients were temporary residents whose medical records were not available).

16.3% of patients who tested positive and 16.7% of those who tested negative returned.Of those testing positive, 38% had antibiotic side effects and 62% had ongoing symptoms.Of those testing negative, 60% had other symptoms of viral URTI and 26% had ongoing sore throat symptoms.One patient was admitted to hospital with Epstein Barr virus and one was seen with unrelated symptoms.

These results are summarised in the figures below (Figure 2 and Figure 3).

## 4. Discussion

The results reported here further support the use of POCT using nucleic acid technology to confirm the presence of symptomatic streptococcal infection after initial assessment using the NICE guidelines. The potential impact of applying this strategy to the management of acute sore throat compared to usual practice is shown in Table 5 below.

While the DEGAM guidelines currently offer the option of a rapid GAS test for 3–15-year-old patients, uptake has been limited due to the poor sensitivity and specificity of previous rapid antigen tests. The development of highly accurate molecular testing has the potential to significantly improve the performance of both the NICE and DEGAM guidelines in reducing unnecessary antimicrobial use while minimising the risk of complications from untreated streptococcal infections. These results are consistent with other studies [16,19] that have previously shown that POCT performs better than the use of Centor scores alone. The potential reduction in antibiotic prescribing reported here is similar to that seen in a retrospective study on the impact of polymerase chain reaction POCT on antibiotic prescribing rates [20], which found a reduction of 44.1% in antibiotic prescribing, suggesting that the results described in our study are likely to be representative of current clinical practice.

Patient and staff surveys previously reported [16] for this QI project suggest high acceptability amongst both groups. Furthermore, the re-consultation rates were the same in patients testing both positive and negative on POCT. The reasons for repeat consultation in patients testing negative support the diagnosis of non-bacterial throat infection, with most reporting other symptoms of viral upper respiratory tract infections, such as cough and coryza. In the group that tested positive on POCT, 38% complained of side effects of antibiotics such as diarrhoea and 62% of ongoing symptoms. This suggests a role for improved patient counselling and safety-netting in both groups of patients. One (1/144) of the patients sampled, who had tested negative, was admitted to hospital within 10 days of being assessed as part of this project; however, a review of the discharge summary confirmed a diagnosis of Epstein-Barr virus infection.

This work also demonstrates the potential for safe and efficient transfer of diagnosis from doctors to non-medical members of primary care teams. This is particularly relevant in the United Kingdom context, where recent strategies to improve access to primary care [21,22] have significantly increased the proportion of consultations for minor illness carried out by community pharmacists, nurses and paramedics. This additional capacity will improve access but has the risk of increasing antibiotic prescribing rates. The ability to obtain a rapid result with high sensitivity and specificity will further enhance the ability of these members of the primary care workforce to manage these patients safely and efficiently, while freeing up doctors to see more complex cases. This conclusion is supported by previous work describing the community pharmacy-led sore throat test and treat service in Wales [23], which identified a 36–47% reduction in antibiotic prescriptions when POCT is used in addition to clinical prediction rules.

Although the addition of point of care testing to usual care comes with increased consumable and set-up costs, it facilitates the transfer of care to lower-paid members of the medical workforce and reduces antibiotic costs. Furthermore, the ability to deliver services outside traditional primary care settings, including in community pharmacies that may be more accessible to patients, suggests a potential role in reducing health inequalities through greater access to prompt diagnostics and treatment.

## 5. Limitations and Future Research Directions

This study estimated the benefits based on a comparison with the gold standard of care as outlined in the NICE guidance [10]. Studies suggest that doctors in particular tend to be less strict in their adherence to guidelines, using their own heuristics based on a number of other factors, such as fear of litigation, desire to satisfy patient expectations and personal perception of risk/benefit [24]. As a result, guidelines are frequently less effective in clinical practice. This suggests that the projected benefits discussed here in terms of the performance against the NICE guidance may underestimate the true benefits of a rapid and accurate point of care test in terms of reducing antibiotic use.

Further work is required to look at the health economics of introducing POCT in terms of the direct care costs and wider system costs, so as to determine the net impact of introducing this technology and the impact on health inequalities. One limitation of this study was the small number of patients in some of the groups, resulting in wide confidence intervals. Larger studies with more patients in each group would improve the statistical accuracy and, in addition, a comparison with usual care process would allow a better assessment of the impact on prescription rates. Finally, these results assume that the reported sensitivity and specificity of the Abbott ID NOW Strep A assay in clinical practice match the reported results. Confirming this with confirmatory microbiological analysis during future studies would add further weight to the arguments for wider use of point of care testing in the management of sore throat.

## 6. Conclusions

This study provides additional support for the potential role of POCT to facilitate targeted antimicrobial prescribing, with significant benefits when compared to current best practice. Further work is required to look at the wider impacts on the health system in terms of costs, access and prescribing data.

## Figures and Tables

**Figure 1 diagnostics-14-02680-f001:**
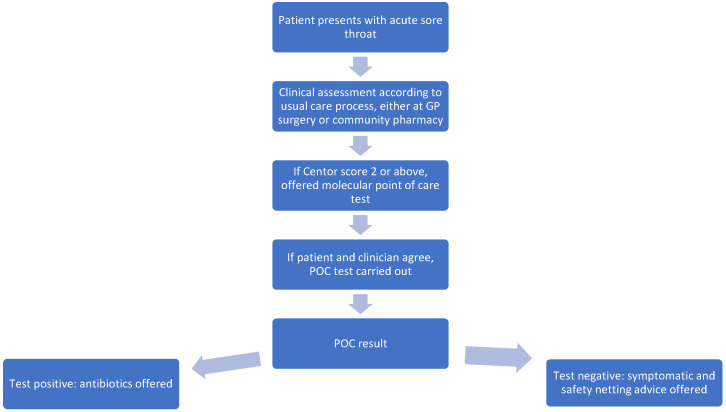
Care process for the initial patient assessment.

**Figure 2 diagnostics-14-02680-f002:**
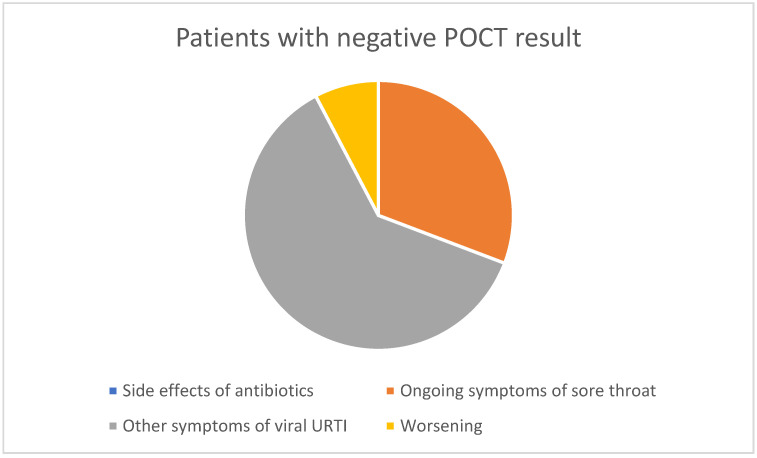
Reasons for re-consultation after a negative POC result.

**Figure 3 diagnostics-14-02680-f003:**
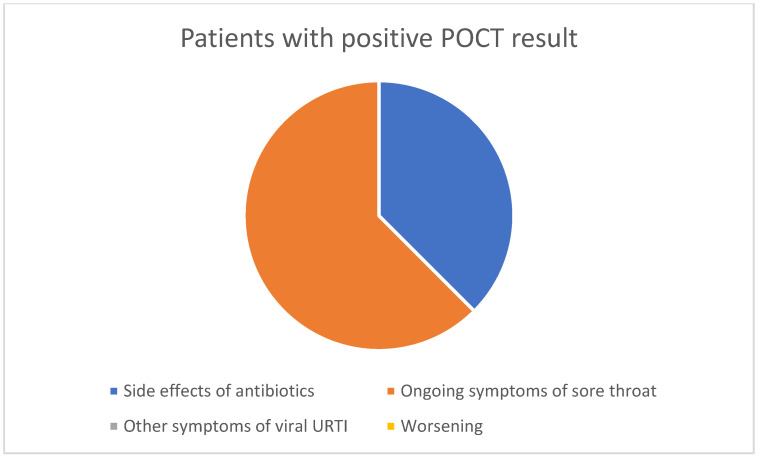
Reasons for re-consultation after a positive POC result.

**Table 1 diagnostics-14-02680-t001:** NICE recommended management of acute sore throat [10].

Clinical Prediction Rule Result	NICE Recommended Management Strategy
**FeverPAIN 0 or 1, Centor 0, 1, 2**	Do not offer antibiotic
**FeverPAIN 2 or 3**	Consider no antibiotic or a back-up prescription
**FeverPAIN 4 or 5, Centor 3 or 4**	Consider an immediate antibiotic or a back-up prescription

**Table 2 diagnostics-14-02680-t002:** DEGAM recommended management of acute sore throat [10].

Clinical Prediction Rule Result	DEGAM Recommended Management Strategy
**FeverPAIN 0–2, Centor 0–2,** **McIsaac 0–2**	Do not offer antibiotic
**FeverPAIN 3,** **Centor 3,** **McIsaac 3**	Age 3–15: Optional GAS rapid test; if positive, immediate or delayed prescription Other patients: immediate or delayed prescription
**FeverPAIN 4 or 5, Centor 4,** **McIsaac 4–5**	Consider an immediate antibiotic or a back-up prescription

**Table 3 diagnostics-14-02680-t003:** FeverPAIN score vs. POCT result.

Fever Pain Score	% Positive on ID NOW (95% Confidence Interval)	*n*
1	26.7 (12.3–45.9)	30
2	25.6 (13.0–42.1)	39
3	44.1 (27.2–62.1)	34
4	66.7 (34.9–90.1)	12
5	50 (11.8–88.2)	6
Total	36.4 (27.8–45.6)	121

**Table 4 diagnostics-14-02680-t004:** McIsaac score vs. POCT result.

McIsaac Score	% Positive on ID NOW (95% Confidence Interval)	*n*
2	14.5 (6.5–26.7)	55
3	45 (29.3–61.5)	40
4	68.2 (45.1–86.1)	22
5	50 (11.8–88.2)	6
Total	35.8 (27.3–44.9)	123

**Table 5 diagnostics-14-02680-t005:** Potential impact on antibiotic prescribing decisions.

Clinical Prediction Rule Result	NICE Recommended Management Strategy	POC Positive (%) Result	Potential Impact on Prescribing Decisions *
**FeverPAIN 0 or 1**	Do not offer antibiotic	26.7% (12.3–45.9)	No impact (CPR does not support prescribing)
**Centor 0, 1, 2**	Do not offer antibiotic	22.7% (13.3–34.7)	No impact (CPR does not support prescribing)
**FeverPAIN 2 or 3**	Consider no antibiotic or a back-up prescription	34.2% (23.5–46.3)	65.8% reduction (assuming back-up antibiotic prescribed)
**Centor 3 or 4**	Consider an immediate antibiotic or a back-up prescription	44.9% (33.6–56.60	55.1% reduction
**FeverPAIN 4 or 5**	Consider an immediate antibiotic or a back-up prescription	61.1% (35.7–82.7)	38.9% reduction

* Assuming all patients opted for antibiotics following an informed shared decision-making process.

## Data Availability

The raw data supporting the conclusions of this article will be made available by the authors on request.

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
