# Peer review of "Improving Antimicrobial Stewardship in Acute Sore Throat: Comparison of FeverPAIN and McIsaac Scores with Molecular Point of Care Testing Using Abbott ID NOW"

_diagnostics, 2024, doi:10.3390/diagnostics14232680_

Round 1
Reviewer 1 Report
Comments and Suggestions for Authors
Thank you for conducting this research. I have few points to be addressed:
1- The introduction has relatively long sentences that make delivering ideas to the reader sub-optimal. I suggest checking the language and punctuation in the whole manuscript.
2- Although this was a retrospective study, you need to justify using the return consultation with patients instead of directly following up within by calling them for example to retrieve information. Why did you assume that patients will return or comeback to the same GP or community pharmacy if they had something to say?
3- I wonder if Table 5 should belong to Results of Discussion ... please explain.
Author Response
Reviewer 1
"The introduction has relatively long sentences that make delivering ideas to the reader sub-optimal. I suggest checking the language and punctuation in the whole manuscript."
Response: this has been rewritten
"Although this was a retrospective study, you need to justify using the return consultation with patients instead of directly following up within by calling them for example to retrieve information. Why did you assume that patients will return or comeback to the same GP or community pharmacy if they had something to say?"
Explanation included in text relating the the medical records systems used in the study population
"I wonder if Table 5 should belong to Results of Discussion ... please explain"
This table relates to the theoretical application of the results to NICE guidance, rather than being a result.
Reviewer 2 Report
Comments and Suggestions for Authors
The study carried out by Rob Daniels and colleagues addresses the important issue of unnecessary antibiotic prescription for acute sore throat. They compare the results of Abbott ID NOW Strep A assay with risk stratification methods based on the FeverPAIN and McIsaac scores and their findings highlight that antibiotics were frequently prescribed unnecessarily across nearly all clinical scores.
In the abstract, there are discrepancies in the values reported in the manuscript. For instance, the abstract mentions 148 patients analysed, while the results section states 144. Similarly, on line 21, the percentage of patients with FeverPAIN scores of 4 or 5 who tested positive via POC is stated as 50-61%, but Table 3 shows a range of 50% to 66.7%.
In addition, the abstract on line 22 should include the McIsaac score value.
On line 29, the keyword "GAS" should be written out in full, followed by the abbreviation in parentheses.
On line 51, it is recommended that the authors replace "sensitivity analysis" with a more precise term.
On line 70, the abbreviation "CPR" should be spelled out.
On line 122, the percentage has been incorrectly rounded to 86%. The actual value is 85.42%, which should be rounded down to 85%.
On line 130, to enhance readability, we suggest the authors include an "and" in the sentence, so it would read as follows: “16.3% of patients who tested positive and 16.7% of those who tested negative returned”
In the caption for Figure 3, we recommend that the authors replace "side effects" with "antibiotic side effects," as the side effects specifically pertain to those associated with taking the antibiotic.
In Table 5, I recommend that the authors modify "POC result" to "POC positive result" for greater clarity.
And in table 5, for the same reason, I suggest the authors put “65.8% reduction”
In the discussion, on lines 159 and 160, I recommend that the authors replace the term "half" with the accurate percentages, as 38% and 62% are not equivalent to "half."
On lines 172-174, the value judgment expressed by the authors (…with clinicians having less clinical and consultation skills training to allow them to conduct what can sometimes be complex consultations with shared decision making and clinical uncertainty.) reflects their personal opinion and lacks scientific evidence to support it. Therefore, the sentence should be revised or omitted.
I recommend that the authors expand the discussion by comparing the study's findings with those reported in existing literature.
Author Response
Reviewer 2
"In the abstract, there are discrepancies in the values reported in the manuscript. For instance, the abstract mentions 148 patients analysed, while the results section states 144. Similarly, on line 21, the percentage of patients with FeverPAIN scores of 4 or 5 who tested positive via POC is stated as 50-61%, but Table 3 shows a range of 50% to 66.7%.
In addition, the abstract on line 22 should include the McIsaac score value.
On line 29, the keyword "GAS" should be written out in full, followed by the abbreviation in parentheses.
On line 51, it is recommended that the authors replace "sensitivity analysis" with a more precise term.
On line 70, the abbreviation "CPR" should be spelled out.
On line 122, the percentage has been incorrectly rounded to 86%. The actual value is 85.42%, which should be rounded down to 85%.
On line 130, to enhance readability, we suggest the authors include an "and" in the sentence, so it would read as follows: “16.3% of patients who tested positive and 16.7% of those who tested negative returned”
All changes made as suggested
"In the caption for Figure 3, we recommend that the authors replace "side effects" with "antibiotic side effects," as the side effects specifically pertain to those associated with taking the antibiotic."
Text amended
"In Table 5, I recommend that the authors modify "POC result" to "POC positive result" for greater clarity.
And in table 5, for the same reason, I suggest the authors put “65.8% reduction”
In the discussion, on lines 159 and 160, I recommend that the authors replace the term "half" with the accurate percentages, as 38% and 62% are not equivalent to "half."
All changes made as suggested
"On lines 172-174, the value judgment expressed by the authors (…with clinicians having less clinical and consultation skills training to allow them to conduct what can sometimes be complex consultations with shared decision making and clinical uncertainty.) reflects their personal opinion and lacks scientific evidence to support it. Therefore, the sentence should be revised or omitted."
Paragraph rewritten
"I recommend that the authors expand the discussion by comparing the study's findings with those reported in existing literature"
Further discussion of reports in literature added.
Round 2
Reviewer 2 Report
Comments and Suggestions for Authors
In the report, under “Recommendations for Authors”, I highlighted the need for improving the description of the methods, and I want to stress one key omission- the absence of a description of the statistical analysis conducted in the materials and methods section.
As noted in my initial review, on line 51, I recommend replacing the term "sensitivity analysis" with a more accurate term, such as “antibiotic susceptibility testing.”
Additionally, as mentioned previously, the discussion section would be enhanced by referencing similar studies already published. This would allow for a comparison and contextualization of the results, such as the studies by:
- Luo R, Sickler J, Vahidnia F, Lee YC, Frogner B, Thompson M. Diagnosis and Management of Group a Streptococcal Pharyngitis in the United States, 2011-2015. BMC Infect Dis. 2019 Feb 26;19(1):193. doi: 10.1186/s12879-019-3835-4.
- May L, Sickler J, Robbins EM, Tang S, Chugh K, Tran N. The Impact of Point-of-Care Polymerase Chain Reaction Testing on Prescribing Practices in Primary Care for Management of Strep A: A Retrospective Before-After Study. Open Forum Infect Dis. 2022 Mar 24;9(5):ofac147. doi: 10.1093/ofid/ofac147.
- Atlas SJ, McDermott SM, Mannone C, Barry MJ. The role of point of care testing for patients with acute pharyngitis. J Gen Intern Med. 2005 Aug;20(8):759-61. doi: 10.1111/j.1525-1497.2005.0143.x.
Author Response
In the report, under “Recommendations for Authors”, I highlighted the need for improving the description of the methods, and I want to stress one key omission- the absence of a description of the statistical analysis conducted in the materials and methods section.
- Thank you - i have added details.
As noted in my initial review, on line 51, I recommend replacing the term "sensitivity analysis" with a more accurate term, such as “antibiotic susceptibility testing.”
2. I have amended text
Additionally, as mentioned previously, the discussion section would be enhanced by referencing similar studies already published. This would allow for a comparison and contextualization of the results, such as the studies by:
3. I have added additional discussion and included the papers suggested, thank you for all your support
Round 3
Reviewer 2 Report
Comments and Suggestions for Authors